# Self-Reporting of Post-Vaccination Symptoms in the COVID-19 Vaccination Process for Teachers in a North Region of Poland

**DOI:** 10.3390/vaccines13101054

**Published:** 2025-10-14

**Authors:** Tadeusz Jędrzejczyk, Anna Tyrańska-Fobke, Agata Konieczna, Daniel Ślęzak, Monika Waśkow, Katarzyna Brzychcy, Piotr Robakowski, Marlena Robakowska

**Affiliations:** 1Department of Public Health & Social Medicine, Faculty of Health Sciences with the Institute of Maritime and Tropical Medicine, Medical University of Gdańsk, 80-210 Gdańsk, Poland; tadeusz.jedrzejczyk@gumed.edu.pl (T.J.); agata.konieczna@gumed.edu.pl (A.K.); marlena.robakowska@gumed.edu.pl (M.R.); 2Department of Medical Rescue, Faculty of Health Sciences with the Institute of Maritime and Tropical Medicine, Medical University of Gdańsk, 80-210 Gdańsk, Poland; daniel.slezak@gumed.edu.pl; 3Institute of Health Sciences, Pomeranian University of Slupsk, 76-200 Slupsk, Poland; monika.waskow@upsl.edu.pl; 4Institute of Management, University of Szczecin, 70-453 Szczecin, Poland; katarzyna.brzychcy@usz.edu.pl; 5Department of Strategic Studies and Security, Institute of Political Science, Faculty of Social Sciences, University of Gdansk, 80-309 Gdansk, Poland; piotr.robakowski@ug.edu.pl

**Keywords:** COVID-19 vaccine, self-reporting, post-vaccination symptoms, occupational group, Poland

## Abstract

**Background:** Poland was one of only 10 European countries listed teachers as a priority group for vaccination against COVID-19 among National Vaccination Program (NVP). The aim of this study was to analyse post-vaccination symptoms self-reported by teachers vaccinated under the national COVID-19 vaccination programme. **Methods:** The presented cross-sectional survey was conducted among teachers from all levels of education in public and non-public institutions, who received the SARS-CoV-2 virus vaccination campaign with the vaccine from AstraZeneca as part of the NVP. The survey was conducted using an original, self-designed questionnaire prepared for this study and distributed to teachers in the form of an online survey via email. Bayesian logistic and linear regression were used to estimate the relationship between predictors and dependent variables. **Results:** A total of 4622 teachers took part in the survey. Of this number, 3908 teachers declared that they had taken the vaccine. (84.5%). In the study group, self-reported late post-vaccination reactions were very strongly [logBF > 3.4] associated with both gender and age. In contrast, self-reporting of serious late post-vaccination symptoms other than fever was very strongly associated only with gender. Only a small proportion of teachers (from 1.45% to 5.34% depending on age and gender) self-reported immediate post-vaccination reaction (up to 15 min after injection). **Conclusions:** Self-reporting of symptoms is a valuable tool for monitoring the effectiveness and safety of vaccinations and can also contribute to increased satisfaction with the vaccination process, especially when patients are made aware that post-vaccination symptoms are a natural sign of the body’s immune response.

## 1. Introduction

Poland was one of only 10 European countries that listed teachers as a priority group for vaccination against COVID-19 [1]. Vaccination of those working in educational institutions lasted from 15 January to 15 April 2021. Teachers and academic staff did not self-register for vaccination; instead, their employers coordinated the process. In the first phase, eligible groups included preschool teachers and aides, primary school teachers (grades I–III), teachers in special education, vocational instructors, pedagogical staff of counselling centres, and institutional managers. From 15 February 2021 (Phase II), eligibility was expanded to all teachers across educational institutions and academic staff [2].

The procedure was standardized: teachers declared willingness to their employer, who submitted the required form via the Educational Information System (SIO) or the Government Safety Centre (for nursery staff). Data were transferred to nodal hospitals, generating e-referrals in the Internet Patient Account (IKP). Employers then arranged vaccination schedules and informed staff, while university employees were registered through the POL-on system [3].

Vaccination centres operated through designated medical facilities, including primary and specialist care, dedicated units, and mobile teams. Initially, qualification was performed only by physicians; however, after additional training, it was extended to other healthcare professionals such as diagnosticians, paramedics, physiotherapists, and medical students [4].

Vaccine safety surveillance was based on the use of current mechanisms and institutions. Surveillance involved a process of control, monitoring, and verification for investigation of suspected quality defects or adverse vaccination reactions. Supervising institutions include the General Pharmaceutical Inspectorate (GIF), the State Sanitary Inspectorate and the Chief Sanitary Inspectorate (GIS), the Military Sanitary Inspectorate, the National Institute of Public Health-National Institute of Hygiene (NIZP-PZH), or the Office for Registration of Medicinal Products, Medical Devices and Biocidal Products (URPL). Real-time data monitoring and long-term observations made it possible not only to assess the efficacy of vaccines, but above all to really evaluate the effectiveness of vaccination [4].

Vaccination aims to increase immunity to a given infectious disease. In the case of COVID-19, we were dealing with a new pathogen, which made it difficult to assess the effectiveness of the vaccination campaign before it was organised. An important element in assessing the effectiveness of the programme should be its impact on changing the dynamics of the epidemic and thus reducing the overload on the healthcare system, in particular hospitals, including intensive care units [5].

Vaccination remains a medical intervention with measurable clinical and, indirectly, social effects. The very system of organising vaccination for a professional group may mean a longer or shorter break from work. In the case of limitations in the number of vaccines themselves, the number of points providing the service, and limitations related to the storage and transport regime of the preparation, it results in the need to devote part or all of the working day to vaccination. Post-vaccination reactions are observed in a significant proportion of vaccinated individuals [6]. Some individuals experience symptoms so severe that they decide to take sick leave, usually for one day.

The decision to take time off work depends on many factors [7] unrelated to the severity of the symptoms. More severe reactions, classified as NOP, are much rarer and have little impact on the assessment of the effects of the vaccination programme [8]. Self-reporting of post-vaccination symptoms is a form of e-health tool use, which was previously difficult and more expensive to implement. Self-reporting is also a potentially useful tool for follow-up after any medical intervention, allowing the identification of health-threatening complications after treatment and possible reactions, as well as feedback on the successful and effective completion of the procedure. In the case of protective vaccination, the usual symptoms are perceived as negative, but in practice they provide information that the immune system has responded positively. Such information is also valuable and can be sent back. Self-reporting is also a tool that offers hope for improving satisfaction with services, providing a sense of continuity of care. Achieving the right level of satisfaction with the service, particularly in terms of communication, may have an indirect impact on shaping the attitudes of pupils and students towards protective vaccinations in general [9].

Previous analyses of the safety of COVID-19 vaccines have been based mainly on data from clinical trials or official reports of adverse reactions following vaccination. Such sources rarely cover the full spectrum of mild and short-term reactions, which are nevertheless important from the perspective of those vaccinated. There is a lack of large studies based on self-reported symptoms by representatives of professional groups other than healthcare workers. Teachers were one of the first professional groups to be vaccinated in Poland, which makes their experiences particularly valuable. Analysis of their reports provides new evidence to supplement pharmacovigilance data, allowing not only for a better assessment of the vaccine’s safety and reactogenicity, but also for an understanding of the impact of post-vaccination reactions on professional functioning and acceptance of the vaccination programme.

Previous studies on COVID-19 vaccine safety and reactogenicity have largely focused on healthcare workers or general population samples. However, there are relatively few analyses specifically addressing occupational groups outside the health sector, such as teachers. A study conducted among Polish educators demonstrated that the ChAdOx1-S vaccine was well tolerated and immunogenic, although side effects were more frequently reported by women and younger teachers [10]. At the regulatory level, the European Medicines Agency (EMA) has consistently emphasized that the benefits of COVID-19 vaccines outweigh the risks, but these evaluations are based mainly on clinical trial data and pharmacovigilance systems [11]. This underlines the need for complementary evidence based on patient self-reporting, which can capture mild and transient post-vaccination reactions typically underrepresented in official registries and situate the experiences of occupational groups, such as teachers, in a broader international context.

The aim of this study was to analyse post-vaccination symptoms self-reported by teachers vaccinated under the national COVID-19 vaccination programme.

## 2. Materials and Methods

A cross-sectional online survey was conducted in April–May 2021 among 4622 teachers from all levels of education in the Pomeranian Voivodeship, representing 14.23% of the regional teacher population [12]. Teachers were selected as the study group as they constituted the second occupational group vaccinated in Poland after healthcare workers, reflecting their high priority in the national vaccination schedule [1].

The study instrument was an original 31-item questionnaire, consisting primarily of closed-ended and Likert-scale questions. A pilot study was carried out with 10 teachers, whose results were excluded from the analysis. The structure of the questions addressed to teachers consisted of several subsets. The section (7 questions) concerning vaccine side effects was based on principles analogous to those used to assess patient experience with healthcare services. The question format involved selecting one of the answers or classifying the severity of symptoms in the context of the respondent’s own experience. In terms of the occurrence of specific effects attributed to vaccination, such as muscle pain, headache or feeling of malaise, respondents were asked to confirm or deny the occurrence of such symptoms. To verify the appropriate level of quality of the questionnaire, a pilot review was conducted on a sample of 10 teachers, 6 of whom represented primary schools and 4 secondary schools. After conducting the electronic survey, responses were verified during a telephone interview. Based on feedback, some questions were reworded.

Data collection followed the snowball method via educational authorities and school directors. Participation was voluntary, anonymous, and uncompensated.

Bayesian logistic and linear regression analyses were conducted using R (version 4.0.2) with the brms package [13]. Model comparisons were based on Bayes Factors (log[BF]) [13], with thresholds of <1.1 (no relationship), 1.1–2.3 (moderate), 2.3–3.4 (strong), and >3.4 (very strong). Ethical approval was granted by the Independent Bioethics Committee of the Medical University of Gdańsk.

## 3. Results

A total of 4622 teachers took part in the survey. Of this number, 3908 teachers declared that they had taken the vaccine (84.5%). The vast majority of the survey group were women (82.63% vs. 14.53%). The predominant age groups were those aged between 41 and 50 years (36.8%). Detailed characteristics of the surveyed population are presented in Table 1.

The surveyed teachers self-reported the occurrence of symptoms in three main groups: immediate post-vaccination reaction (up to 15 min after injection), late post-vaccination reactions and more severe post-vaccination symptoms than fever. When asked about symptoms observed later than immediately after the injection, the participants were asked to specify the time of their appearance in hours. On average, it was 12 h after the injection. In the study group, self-reported late post-vaccination reactions were very strongly [logBF > 3.4] associated with both gender and age. In contrast, self-reporting of serious late post-vaccination symptoms other than fever was very strongly associated only with gender. Only a small proportion of teachers (from 1.45% to 5.34% depending on age and gender) self-reported immediate post-vaccination reaction (up to 15 min after injection). In contrast, most of respondents (ranging from 55.34% to 77.22% depending on age and gender) self-reported late post-vaccination reactions. In addition, also up to a third of respondents (from 18.95% to 37.09% depending on age and gender) reported of more severe post-vaccination symptoms than fever. Detailed results of the types of post-vaccination symptoms reported by the teachers are presented in Table 2.

In the study group of vaccinated teachers, the most frequently reported late post-vaccination symptom was fever. Self-reported occurrence of fever was very strongly [logBF > 3.4] associated with both age and gender. The vast majority of respondents (from 29.32% to 55.54% depending on age and gender) self-reported fever above 38 degrees C. Detailed data on self-reported increases in body temperature as a late post-vaccination symptom are presented in Table 3.

The incidence of the remaining late post-vaccination symptoms self-reported by the respondents can be assessed from the data in Table 4. Table 4 presents data on the remaining symptoms reported by the respondents. Respondents were able to self-report the following late post-vaccination symptoms other than fever: musculoskeletal pains, headaches, feeling tired/weak/broken, shivers, concentration difficulties and feeling irritable. In most of the aforementioned symptoms other than fever, their self-reporting was very strongly related [logBF > 3.4] to age. Only in the case of feeling tired/weak/broken was there a strong [logBF 2.3–3.4] association also with the gender of the respondents. Feeling tired/weak/broken was the most frequently self-reported symptom in group of man. The occurrence of this symptom was self-reported by 1.91% to even 14.85% of respondents depending on age and gender. In contrast, the most frequently self-reported symptom that was present in respondents was feeling irritable. Its presence was self-reported by 24% to 43.55% of respondents. Details of other post-vaccination symptoms can be found in Table 4.

The intensity of individual late post-vaccination symptoms is presented in Appendix A, which can be found in the Appendix A to the manuscript. Respondents self-reported the subjectively perceived intensity of post-vaccination symptoms compared to the symptoms they experienced at the time of their last infection. The intensity of only some symptoms showed a relationship with the demographic characteristics of the subjects. The intensity of musculoskeletal pains was very strongly [logBF > 3.4] associated with age. In contrast, headache intensity scores were very strongly associated with gender and related [logBF > 1.1] to age. Feeling tired/weak/broken and concentration difficulties were very strongly associated with age. Most of the surveyed teachers self-reported that the post-vaccination symptoms they felt, such as musculoskeletal pains, headaches, feeling tired/weak/broken and concentration difficulties were stronger than during their last infection. In contrast, shivers were, according to the most of respondents, significantly stronger than during the last infection.

## 4. Discussion

The study presents a wide range of post-vaccination symptoms that were self-reported by the study population of teachers vaccinated against COVID-19 in northern Poland. The self-reported symptoms experienced by the study population are consistent with those described in the international literature. A study conducted by Dreyer and colleagues in 2022 found that 92.4% of participants reported at least one adverse reaction after vaccination against COVID-19. The most common symptoms were pain at the injection site (78.9%), fatigue (70.3%) and headache (49.0%). These symptoms were usually mild and resolved within a few days [14]. A review also conducted in 2022 by SeyedAlinaghi and colleagues discussed rare but serious adverse reactions following COVID-19 vaccination, such as allergic reactions, neurological disorders, and cardiovascular complications, including myocarditis and pericarditis. It was pointed out that these reactions are rare and their risk is significantly lower than the benefits of vaccination [15]. In contrast to these international studies that often aggregate data from mixed populations, our analysis is one of the few that focuses specifically on teachers, a professional group prioritised early in national vaccination schedules. This occupational focus is crucial, as teachers’ absences due to vaccine side effects can directly affect the functioning of schools and continuity of education—an aspect rarely captured in broader surveillance systems. Unlike our findings, these international analyses were not able to quantify the occupational impact of symptoms, which is a central contribution of the present study. However, a study conducted by Amer and colleagues in 2024 documents the occurrence of symptoms suggestive of myocarditis and pericarditis after COVID-19 vaccination, such as chest pain (88 cases), shortness of breath (103 cases), and a feeling of rapid heartbeat (34 cases). These symptoms were consistent with the CDC report on these conditions [16]. Furthermore, a study conducted by Li et al. (2024) found that COVID-19 vaccination is safe in individuals with a history of allergic reactions, provided that they are adequately monitored after vaccination. No increased risk of anaphylactic reaction was observed in these patients [17]. Our findings are consistent with recent systematic reviews confirming that most post-vaccination symptoms are mild and self-limiting, while serious adverse events remain rare [18,19]. Importantly, our study also provides insights into the occupational consequences of vaccine reactogenicity. The proportion of teachers reporting temporary absence from work due to side effects aligns with global evidence indicating that vaccine-related absenteeism among employees is not negligible. A recent meta-analysis estimated that approximately 17% of healthcare workers experienced absenteeism related to vaccination side effects [20], while comparable studies in other occupational groups reported similar patterns [21]. While the proportion of our respondents reporting work absence (27%) is somewhat higher than the pooled prevalence of 17% among healthcare workers [20], this difference may reflect both occupational context and timing. Unlike healthcare workers, many teachers could not easily reschedule duties or rely on shift replacements, which likely increased the impact of even short-lived symptoms on absenteeism. This suggests that occupation-specific dynamics should be taken into account in planning vaccination campaigns, underscoring the added value of teacher-focused data. Taken together, these results highlight that although vaccine reactogenicity is generally manageable, its short-term impact on workforce availability should be considered when planning large-scale immunisation campaigns targeting specific professional groups. In this respect, our study adds valuable evidence by documenting the unique experience of teachers, a group seldom addressed in international literature. Self-reported post-vaccination symptoms are particularly important because they provide insight into the body’s immune response. In the case of viral vector vaccines such as ChAdOx1-S (AstraZeneca), studies consistently indicate that most adverse reactions are mild to moderate, short-lived, and occur more frequently after the first dose [18,22]. A study conducted among Polish teachers demonstrated both good tolerability and strong immunogenicity of ChAdOx1-S, with side effects more commonly reported by women and younger participants [10]. Similarly, large multinational cohort analyses confirmed that while local and systemic symptoms such as fever, headache, or fatigue are common after AstraZeneca vaccination, severe adverse events remain rare and their risk is outweighed by the benefits of immunization [11,19]. Taken together, these findings suggest that reactogenicity, although sometimes inconvenient for recipients, may serve as a marker of an effective immune response while also highlighting the importance of monitoring occupational impacts such as short-term absenteeism. Our results therefore complement existing international evidence by providing large-scale data from teachers, a professional group seldom analysed in studies of AstraZeneca vaccination. Previous analyses of ChAdOx1-S immunogenicity in Polish teachers [10] relied on serological testing, while our study demonstrates that subjective symptom reporting can complement such biomedical measures by reflecting the lived experiences of vaccine recipients. Compared to EMA pharmacovigilance data, which emphasise rare severe events, our findings document the frequency and perceived intensity of common reactions, thus filling an important surveillance gap.

Furthermore, a study conducted among healthcare workers showed that self-reported post-vaccination reactions are common and can be used to monitor vaccination safety and as an educational tool to inform about an effective immune response [23]. A Stanford University study also found that individuals who interpret post-vaccination symptoms as a sign of a positive immune response report higher health satisfaction and stronger immune responses. Making patients aware that symptoms such as fever and fatigue are natural can increase confidence in vaccination [24].

The tool for patients to self-report symptoms they experience in various health circumstances is also widely described in the international literature. A systematic review of the literature assessed the reliability of self-reporting by employees of work-related illnesses. The results indicate that self-reporting can provide valuable information on the presence of diseases, especially in the case of musculoskeletal and skin disorders. However, overall agreement with expert assessments was low to moderate, suggesting a need for further research and supplementation of self-reporting with clinical examinations [25].

The issue of self-reporting has also been analysed in the context of surgical treatment. One study showed that optimising methods of self-reporting surgical complications, including the use of standardised electronic platforms and rigorous assessment by medical staff, leads to improved quality of care and reduced complication rates [26].

In contrast, a study conducted among primary care patients assessed differences in self-reported sick leave days after a brief intervention aimed at preventing work-related stress-related sick leave. Although the intervention did not show significant differences in the number of sick days, the use of self-reporting made it possible to take into account short-term absences that are not recorded in official data [27].

The accuracy of the self-reporting tool is also important. In the case of healthcare utilisation assessments, a study comparing self-reported data with administrative data showed greater consistency for monthly than for annual healthcare utilisation indicators. This suggests that self-reporting can be a reliable source of feedback on the implementation of medical procedures over shorter periods of time [28].

It is also worth noting a systematic review of studies that assessed the impact of patient feedback on the effectiveness of medical procedures. The results suggest that feedback can lead to changes in clinical practices, improved doctor–patient communication and increased patient satisfaction [29].

Unlike self-reporting in surgical care or occupational disease surveillance, where discrepancies with expert assessment often limit interpretability, in the vaccination context self-reports provide unique and otherwise unavailable information. By capturing short-term and transient symptoms that rarely reach clinical attention, our data help situate vaccine safety not only in medical but also in social and occupational terms. Currently, various e-health solutions are identified as effective self-reporting tools. One systematic review of studies assessed the impact of e-health tools on patients’ self-reporting of adverse events and symptoms. The results suggest that well-functioning e-health tools can help patients better understand their health and lead to improved patient–doctor relationships. E-health tools as useful solutions for self-reporting are also being analysed in other contexts of health behaviours and conditions [30]. For example, one study evaluated the acceptability, feasibility, and usability of a digital tool for self-reporting symptoms of anxiety and perinatal depression. The results suggest that such tools can be effective in identifying patients’ needs and referring them to appropriate services [31]. Another pilot study assessed the feasibility of an e-health intervention aimed at improving health-related quality of life in patients with chronic pain. The results indicate that the intervention was well received by patients and has potential for further development [32]. In contrast, a qualitative study conducted among patients and general practitioners identified needs related to self-management of persistent physical symptoms. The results indicate the need for early recognition of such symptoms and support for patients in their self-management, which can be achieved through e-health tools [33].

It is also worth noting the cost-effectiveness of various e-health solutions, including those used for self-reporting of various information by patients. A systematic review of studies evaluating the cost-effectiveness of digital health interventions has shown that many of them are cost-effective, offering better quality of life at lower costs compared to traditional methods [34].

In contrast, our teacher-focused dataset demonstrates how self-reporting can be applied in a large-scale vaccination campaign, capturing outcomes that are both clinically relevant and socially meaningful.

Overall, our study extends the international literature by providing large-scale, occupation-specific evidence on AstraZeneca vaccination. It highlights that while patterns of reactogenicity are broadly consistent with other reports, the consequences for work absence and professional functioning are context-dependent. These insights are directly relevant for vaccine safety surveillance, as they underscore the value of integrating self-reported outcomes from diverse occupational groups into broader monitoring systems. One of the most important limitations of the presented study is the fact that the group of teachers surveyed may not be fully representative of this professional group in the Pomeranian Province. Participation was voluntary, and the use of an online survey distributed through schools and educational authorities introduces the possibility of self-selection bias. Teachers who experienced stronger post-vaccination reactions may have been more motivated to participate, which could lead to overreporting of symptoms. Conversely, individuals with mild or no symptoms may have been less likely to complete the survey, resulting in potential underreporting. Second, due to restrictions related to the epidemiological situation, it was not possible to conduct the survey directly at vaccination sites, which further limited the representativeness of the sample. Third, the use of a proprietary questionnaire with predefined response categories may have introduced a certain degree of reporting bias by suggesting specific symptoms to respondents.

## 5. Conclusions

The findings of the study confirm that post-vaccination symptoms following COVID-19 vaccination among teachers were common but usually mild and short-lived. The most commonly reported symptoms were fever, muscle aches and fatigue, which rarely led to prolonged absences from work. Self-reporting of symptoms is a valuable tool for monitoring the effectiveness and safety of vaccinations and can also contribute to increased satisfaction with the vaccination process, especially when patients are made aware that post-vaccination symptoms are a natural sign of the body’s immune response. Nevertheless, the results of the study should be interpreted with certain methodological limitations in mind, including the specific nature of self-reporting and the representativeness of the study group. In the future, it would be worthwhile to continue research on the long-term effects of vaccination, taking into account various professional and demographic groups.

## Figures and Tables

**Table 1 vaccines-13-01054-t001:** Characteristics of the study population.

Category	Feature	Number	% of 3908
**Gender**	Female	3229	82.63
	Male	568	14.53
	I don’t want to answer that question	111	2.84
	Total	3908	100%
**Age (in years)**	20–30	216	5.53%
	31–40	715	18.3%
	41–50	1438	36.8%
	51–60	1274	32.6%
	61–70	256	6.55%
	Total	3899 *	99.78%
**Have you been forced to give up work because of an increased post-vaccination reaction?**	Yes	1056	27.02%
	No, but only because I managed to get vaccinated before the weekend	907	23.21%
	No, symptoms were not so severe that I had to stay at home	640	16.38%
	No, due to the possibility of remote work	26	0.67%
	No answer	1279	32.72%
	Total	3908	100%

* missing data.

**Table 2 vaccines-13-01054-t002:** Types of post-vaccination symptoms reported by the teachers surveyed.

Feature	Reporting of Immediate Post-Vaccination Reaction (up to 15 Min After Injection)			Reporting of Late Post-Vaccination Reactions			Reporting of More Severe Post-Vaccination Symptoms than Fever		
	%	95% BCI	Log [BF]	%	95% BCI	Log [BF]	%	95% BCI	Log [BF]
**Female**	3.77	2.93–4.7	**−7.00**	70.49	68.51–72.41	**4.7**	37.09	34.43–39.59	**13.9**
**Male**	2.84	1.69–4.43		58.96	54.55–63.14		19.46	15.38–23.89	
**I don’t want to answer that question**	3.81	1.32–8.06		69.23	59.75–77.17		25.24	16.61–35.48	
**20–30** **years**	5.34	2.99–8.75	**0.41**	77.22	71.69–81.75	**15**	28.5	22.52–35.37	**−** **0.97**
**31–40** **years**	5.24	3.55–7.27		71.45	67.73–74.81		31.13	26.6–35.69	
**41–50** **years**	4.21	2.77–5.8		65.66	61.97–69.46		30.07	25.79–34.58	
**51–60** **years**	2.75	1.75–3.95		60.22	56.54–64.17		25.74	21.75–29.62	
**61–70** **years**	1.45	0.68–2.93		55.34	49.19–61.27		18.95	14.19–24.48	

**Table 3 vaccines-13-01054-t003:** Reporting of raised body temperature as a late post-vaccination symptom.

Feature	Up to 37 Degrees C		Between 37.1 and 37.5 Degrees C		Between 37.6 and 38 Degrees C		Above 38 Degrees C		log [BF]
	%	95% BCI	%	95% BCI	%	95% BCI	%	95% BCI	
**Female**	14.16	12.74–15.87	17.46	15.88–19.13	27.21	25.52–29.04	41.15	38.53–43.61	**3.7**
**Male**	21.71	18.29–25.57	21.99	19.75–24.31	26.83	24.95–28.73	29.32	25.19–33.97	
**I don’t want to answer that question**	10.29	6.92–15.28	14.05	10.3–18.2	25.48	21.79–28.27	50.1	38.98–60.76	
**20–30** **years**	8.48	6.47–11.04	12.08	9.66–14.95	23.86	21.01–26.46	55.54	48.17–62.34	**15.8**
**31–40** **years**	13.15	10.98–15.56	16.58	14.46–18.71	26.91	25.1–28.85	43.28	38.68–47.9	
**41–50** **years**	17.21	14.62–19.92	19.55	17.37–21.64	27.44	25.74–29.33	35.75	31.83–40.04	
**51–60** **years**	19.29	16.4–22.64	20.79	18.65–23.00	27.30	25.53–29.13	32.58	28.71–36.6	
**61–70** **years**	18.68	14.9–23.77	20.43	17.73–23.26	27.29	25.46–29.17	33.31	27.52–39.75	

**Table 4 vaccines-13-01054-t004:** Remaining symptoms reported by the respondents.

Feature	%	95% BCI	Log [BF]	%	95% BCI	Log [BF]	%	95% BCI	Log [BF]
	Musculoskeletal Pains			Headaches			Feeling Tired/Weak/Broken		
**Female**	17.73	15,8–19,96	**−** **4.77**	18.59	16.66–20.73	**−** **4.96**	4.26	3.19–5.47	**3.33**
**Male**	24.38	19.86–29.97		24.37	19.72–29.42		10.62	7.51–14.8	
**I don’t want to answer that question**	14.71	7.7–24.51		26.54	17.27–37.71		5.33	1.81–11.69	
**20–30** **years**	13.92	9.82–19.54	**6.26**	17.2	12.14–23.39	**−** **1.00**	1.91	0.76–4.05	**10.93**
**31–40** **years**	14.04	11.18–17.86		19.65	16.28–23.33		3.94	2.46–5.83	
**41–50** **years**	16.34	13.03–20.11		22.67	19.3–26.57		7.09	4.83–9.63	
**51–60** **years**	21.45	17.29–25.78		26.4	22.31–30.52		11.01	7.7–14.65	
**61–70** **years**	30.7	23.73–38.81		30.9	23.92–38.36		14.85	9.28–22.29	
	**Shivers**			**Concentration difficulties**			**Feeling irritable**		
**Female**	18.92	16.83–21.07	**−** **3.19**	23.46	21.21–25.9	**−** **5.58**	31.52	29.09–34.09	**−** **8.5**
**Male**	27.23	22.55–32.43		30.14	25.06–35.68		34.82	29.83–40.34	
**I don’t want to answer that question**	21.23	12.95–31.72		27.65	18.29–38.37		32.21	22.36–43.15	
**20–30** **years**	14.73	10.34–20.35	**7.93**	17.88	12.76–23.61	**6.41**	24.00	18.54–30.2	**5.05**
**31–40** **years**	17.36	14.13–21.22		22.54	18.75–26.44		27.92	23.89–32.15	
**41–50** **years**	21.3	17.87–25.38		27.45	23.48–31.52		32.61	28.49–30.78	
**51–60** **years**	26.94	22.82–31.71		32.47	28.06–36.74		37.85	33.16–42.42	
**61–70** **years**	34.74	27.16–42.3		37.53	29.91–44,85		43.55	36.42–51.1	

## Data Availability

The data is unavailable due to privacy or ethical restrictions.

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
