# Peer review of "Self-Reporting of Post-Vaccination Symptoms in the COVID-19 Vaccination Process for Teachers in a North Region of Poland"

_vaccines, 2025, doi:10.3390/vaccines13101054_

Round 1
Reviewer 1 Report
Comments and Suggestions for Authors
This manuscript presents findings from a large cross-sectional survey of Polish teachers regarding self-reported post-vaccination symptoms following COVID-19 vaccination with AstraZeneca. The topic is relevant to the readership of Vaccines, and the study adds value by focusing on an occupational group prioritized in the national vaccination program. However, several aspects of the study design, reporting, and discussion require clarification and strengthening before the manuscript can be considered for publication
- The introduction is informative regarding the vaccination program in Poland but does not clearly state the knowledge gap addressed by this study. The authors should emphasize why teacher self-reporting adds new evidence compared to existing studies on vaccine safety and reactogenicity
- The questionnaire is described as “self-designed” but no details are provided about validation or pre-testing, apart from a pilot with 10 teachers. More information on questionnaire development, reliability, and validity is needed.
- The results section is detailed but sometimes overwhelming. The presentation could be streamlined, with greater focus on the most relevant findings rather than listing multiple tables with overlapping information.
- Table 1 includes symbols (*, **) without clear footnotes or explanations. This needs correction.
- Some of the percentages and totals (e.g., Table 1, gender categories) do not fully align. Please check internal consistency.
- Although Bayesian logistic/linear regression is described in the Methods, the corresponding results are not clearly presented. The manuscript only reports descriptive percentages with occasional log[BF] annotations, but without regression coefficients, odds ratios, or credible intervals. To support the claim of multivariable associations, full regression results should be presented in a separate table or in the text.
- The discussion extensively cites international literature but often summarizes rather than critically compares findings with the current study. The authors should better contextualize how their results contribute to existing knowledge on vaccine safety surveillance.
- The strengths and limitations section should be expanded. In particular, self-selection bias and the potential for overreporting or underreporting symptoms through self-report need stronger emphasis.
- The reference list is generally appropriate; however, the manuscript would benefit from the inclusion of more recent and directly relevant studies. In particular, comparative data on reactogenicity and absenteeism in occupational groups (e.g., Politis et al., 2024; Abera et al., 2024), systematic reviews on vaccine safety and adverse events (e.g., Yaamika et al., 2023; Faksova et al., 2024), and recent reports from regulatory authorities (EMA assessment reports) should be incorporated. Additionally, studies focusing specifically on teachers or similar populations (e.g., Ganczak et al., 2022) would strengthen both the Introduction and Discussion by situating the findings within the broader international evidence
Reviewer 2 Report
Comments and Suggestions for Authors
This study analyzed self-reported post-vaccination symptoms among 3,908 Polish teachers who received the AstraZeneca COVID-19 vaccine. Most respondents reported late-onset reactions like fever, muscle pain, and fatigue, which were strongly associated with gender and age. Immediate reactions were rare. While common, symptoms were typically mild and short-lived, infrequently causing work absenteeism. The research concludes that self-reporting is a valuable tool for monitoring vaccine safety and effectiveness. Informing individuals that these symptoms signify a natural immune response can increase satisfaction with and confidence in the vaccination process. While this manuscript presents interesting and valuable data, I have a few concerns about the manuscript.
1. The abstract and introduction specify that the study was for teachers vaccinated with the AstraZeneca vaccine. However, the discussion (Page 8) cites literature about the Pfizer-BioNTech (BNT162b2) mRNA vaccine [16, 17]. The immunogenic profiles and common side effects of viral vector (AstraZeneca) and mRNA (Pfizer/Moderna) vaccines are known to differ. The discussion should primarily compare findings to studies on the AstraZeneca vaccine or state that comparisons are made to the broader COVID-19 vaccine literature despite platform differences.
2. The study defines "immediate" as "up to 15 minutes after injection." This is a very short window typically used for monitoring anaphylaxis. Most common vaccine reactions (e.g., pain at the injection site) are not expected within 15 minutes. The authors should justify this specific time frame or consider if a longer period (e.g., first 2-6 hours) would be more relevant for capturing early-onset symptoms.
3. Data in Table 4 is presented as the percentage of people who did NOT experience a symptom. This is the opposite of how the other tables (2 and 3) are presented. Could you please explain the purpose of this data presentation? The authors must ensure the presentation is consistent and clearly labeled to avoid misinterpretation. The very high percentages (e.g., 95.74% for females "not reporting" feeling tired) actually indicate that the symptom was very common, which is a key finding that is obscured by the presentation.
Round 2
Reviewer 1 Report
Comments and Suggestions for Authors
Authors have addressed appropriately the comments.